# An inter-comparison study on the impact of atmospheric boundary layer height on gigawatt-scale wind plant performance

Stefan Ivanell<sup>1</sup>, Warit Chanprasert<sup>1</sup>, Luca Lanzilao<sup>2</sup>, James Bleeg<sup>3</sup>, Johan Meyers<sup>2</sup>, Antoine Mathieu<sup>4,5</sup>, Søren Juhl Andersen<sup>6</sup>, Rem-Sophia Mouradi<sup>4,5</sup>, Eric Dupont<sup>4,5</sup>, Hugo Olivares-Espinosa<sup>1</sup>, and Niels Troldborg<sup>7</sup>

<sup>1</sup>Uppsala University, Department of Earth Sciences, Wind Energy Division, 621 67 Visby, Sweden
<sup>2</sup>KU Leuven, Department of Mechanical Engineering, Celestijnenlaan 300 – box 2421, B-3001 Leuven, Belgium
<sup>3</sup>DNV, One Linear Park, Avon St. Temple Quay, Bristol BS2 0PS
<sup>4</sup>EDF R&D, 6 Quai Watier,78400 Chatou,France
<sup>5</sup>CEREA, École des Ponts, Île-de-France
<sup>6</sup>DTU, Department of Wind and Energy Systems, Koppels Allé 403, 2800 Kgs Lyngby, Denmark
<sup>7</sup>DTU, Department of Wind and Energy Systems, Frederiksborgvej 399, 4000 Roskilde, Denmark

Correspondence: Stefan Ivanell (stefan.ivanell@geo.uu.se)

# Abstract.

The height of the atmospheric boundary layer (ABL) exerts a significant influence on flow behavior within wind farms and directly impacts their performance. This study investigates how variations in ABL height and capping inversion layer thickness affect the efficiency and power output of a gigawatt-scale wind farm. Five advanced numerical approaches, ranging from

- high-fidelity large-eddy simulations (LES) to Reynolds-averaged Navier-Stokes (RANS), are used to model farm-scale flow dynamics under shallow ( $\sim 150m$ ) and deep ( $\sim 500m$ ) ABL conditions. The results consistently show that shallow ABLs increase flow blockage and turbine wake interactions, leading to reduced power production. In contrast, deeper ABLs promote enhanced wake recovery and increased overall energy yield. These trends were observed across all solvers, demonstrating the robustness of the findings. Notably, while some quantitative differences emerged depending on modeling fidelity and com-
- putational domain size, the overarching trends remained consistent among the participating research institutions and industry partners. The study concludes that the sensitivity to model type is limited and that ABL height and stability are critical parameters to consider in wind energy siting and turbine layout design to optimize performance across varying atmospheric conditions.

# 1 Introduction

The interaction between atmospheric winds and utility-scale wind turbines is becoming more complex as the height and rotor diameter of modern turbines increase, especially for an offshore site (Veers et al., 2019). When these turbines are clustered together into farms, the interaction with the atmosphere and atmospheric boundary layer (ABL) becomes even more intricate.

The atmospheric boundary layer (ABL) is the region in the troposphere closest to the ground, in which the flow is experiencing frictional forces due to interactions with the Earth's surface. The ABL is a highly turbulent flow region, and although

- various definitions exist, its height is usually identified using the location above which turbulent stresses disappear. In neutrally and unstably stratified ABLs, the turbulent region is typically capped by a strong temperature inversion (a region in which the potential temperature increases significantly over a few hundred meters), also known as a capping inversion (Shaw et al., 2022). In stable boundary layers, a residual non-turbulent neutral layer may exist between the top of the turbulent boundary layer and the capping inversion. Both capping inversion, as well as stable stratification in the free atmosphere above (driven by global circulation), can have a significant impact on wind farm performance (Smith, 2010; Allaerts and Meyers, 2017, 2018a).
- In the current study, we present an inter-comparison study that investigates the effect of the height of this capping inversion on wind farms. We do this for a set of conventionally neutral boundary layers (with conditions derived from Lanzilao and Meyers, 2024), so that the height of the boundary layer effectively coincides with the height of the capping inversion.
- Wind farm performance is influenced by wake and blockage effects. Wake effects have been extensively studied for many
  years using both numerical and experimental methods (Porté-Agel et al., 2020). Research on wind farm blockage is much more
  recent, and has been largely triggered by field observations reported in Bleeg et al. (2018). In this study, a significant slowdown was observed upstream of a series of wind farms by comparing pre and post construction measurements from available
  met masts, suggesting that the wind farm as a whole is blocking the flow. Two main root causes have been investigated to
  explain this blockage effect. A first set of studies have tried to explain blockage as a purely hydrodynamic effect resulting from
- the cumulative induction of all turbines in the farm (see, e.g., Meyer Forsting et al., 2023, and references therein). A second set of studies have associated blockage with the presence of a capping inversion and lighter air in the free atmosphere above, with perturbations of the height of the boundary layer by the wind farm leading to hydrostatic changes of the pressure in the boundary layer, and the excitation of gravity waves on the inversion layer and in the free atmosphere above (Smith, 2010; Allaerts and Meyers, 2017, 2018a). Recently, Lanzilao and Meyers (2024, 2022) managed to separate both effects, showing
- for a range of existing atmospheric conditions over the North Sea, that the hydrostatic blockage effect is an order magnitude larger than the hydrodynamic component (Lanzilao and Meyers, 2024), though both in principle co-exist in the presence of a capping inversion and free-atmosphere stratification. The stratification not only enhances the adverse pressure gradients and associated wind speed decreases upstream of a wind farm, it also, in turn, increases the pressure drop from the front to the back of the wind farm, enhancing wake recovery and influencing turbine power production throughout the array (Lanzilao and
- Meyers, 2024).

With the recognition of the importance of free-atmosphere stratification for wind farm flows, and the challenges that arise in correctly predicting the pressure field, which is tightly linked to the excitation of gravity waves and a correct set-up of boundary conditions in simulations (Lanzilao and Meyers, 2023), it is of interest to compare the performance of widely used numerical solvers among the wind industry and researchers for wind farm flow cases that are subject to significant hydrostatic effects and

50 gravity waves. In the current study, we compare five such solvers, three that are using a large-eddy simulation framework, and two that are using Reynolds-averaged Navier-Stokes simulation framework. We consider a fixed, densely spaced, wind farm (in which blockage effects are expected to be high), and compare the performance of the different simulation tools for two different ABL (/capping inversion) heights, next to also looking at the effect of the capping inversion thickness.

# 2 Numerical Setup

- In this section, an overview of the simulation cases and the numerical setup for different solvers is presented. Conventionally Neutral Atmospheric Boundary Layers (CNBL) with different BLHs are considered in this study. The boundary layer initialization follows Lanzilao and Meyers (2024) where the initial velocity and potential temperature profiles are generated using the Zilitinkevich (1989) and Rampanelli and Zardi (2004) models, respectively. The Geostrophic wind is set to 10 m/s with a surface roughness ( $z_0$ ) of  $1 \times 10^{-4}$  m. The surface heat flux at the bottom surface is zero according to the CNBL definition.
- The BLHs of 150 and 500 m are investigated. These heights are prescribed by the capping inversion height with a strength (Δθ) of 5 K. Moreover, two different capping-inversion thickness ΔH values are considered, i.e. 100 and 500 m, for the BLH of 500 m. A free lapse rate (Γ) of 4 K/km is applied above the inversion layer. The latitude is set to 55.0°, which represents the latitude of the Doggers Bank offshore wind farm in the North Sea. The parameters for each case are summarized in Table 1. It is noted that Danmark Tekniske Universitet (DTU) and Uppsala Universitet (UU) did not perform the simulation for the H500-dh500 case.

Table 1. Case Definition Summary

| Case       | H [m] | $\Delta \theta$ [K] | $\Delta H[m]$ | Γ [K/km] |
|------------|-------|---------------------|---------------|----------|
| H150       | 150   | 5                   | 100           | 4        |
| H500       | 500   | 5                   | 100           | 4        |
| H500-dh500 | 500   | 5                   | 500           | 4        |

The wind farm consists of 100 IEA 15 MW reference turbine (Gaertner et al., 2020) arranged in a  $10 \times 10$  staggered layout with 5D spacing in both streamwise and spanwise directions as shown in Fig. 1 resulting in a farm length and width of  $L_x^f = 10.8$  and  $L_y^f = 11.4$  km where the x, y and z axes refer to the streamwise, spanwise and vertical directions, respectively. The turbine has a rotor diameter (D) of 240 m and a hub height (HH) of 150 m.

There are five participants from both industry and academia including DNV, Danmarks Tekniske Universitet (DTU), Electricité de France (EDF), Katholieke Universiteit Leuven (KUL) and Uppsala Universitet (UU). The name and type of numerical solvers for each institution are listed in Table 2.

Table 2. List of Participants

| Partner | Model Type | Solver Name  | Turbine Modelling             |
|---------|------------|--------------|-------------------------------|
|         | PANS       | STAR-CCM+    | Rotating ADM                  |
| DTU     | LES        | EllipSvs3D   | Rotating ADM coupled to HAWC2 |
| EDF     | RANS       | code_saturne | Non-rotating ADM              |
| KUL     | LES        | SP-Wind      | Non-rotating ADM              |
| UU      | LES        | SOWFA        | Rotating ADM                  |
|         |            |              |                               |

Figure 1. The layout of an idealized wind farm used in this study. Turbines are marked with a letter T and the subscript numbers indicate the row and column, respectively. The x-axis refers to the streamwise direction.

Statistical calculations for the turbulent flow and turbine output of the transient flow solvers are conducted over a physical simulation period of at least one hour.

- The details of numerical setup for each solver, such as computational domain and mesh resolutions, boundary conditions, numerical schemes and turbine modeling, are provided in the following subsections.

# 2.1 DNV STAR-CCM+ Setup

STAR-CCM+ is a general purpose simulation software package best known for computational fluid dynamics. Within STAR-CCM+, DNV customized a steady-state RANS model for simulation for wind farm flows. The turbulence closure is standard
k-epsilon with modified coefficients. The direct influence of buoyancy on the mean flow is simulated via a shallow Boussinesq formulation; extra terms in the closure equations represent the influence of buoyancy on turbulence. Coriolis terms are in the momentum equation. The turbines are represented with a simple actuator disk model where the body forces are functions of the average axial-component of velocity across the disk. These functions derive from the IEA 15 MW power and thrust curves, defined as functions of hub-height freestream wind speed, using the procedure described in Bleeg and Montavon (2022). More

information on this flow model may be found in Bleeg et al. (2018).

The simulations in this study were run within a domain of size 66 km x 66 km x 17 km. The wind farm is located 40 km downstream of the inflow boundary. The mesh spacing is 12 m around each actuator disk and 24 m around the wind farm.

Vertical inflow profiles are generated using a steady-state 1D, single column precursor simulation with the input potential temperature profile frozen. After the steady-state simulation converges, the potential temperature is unfrozen and the 1D 90 solution is marched in time to confirm the full set of profiles are in quasi-equilibrium.

# 2.2 DTU EllipSys3D Setup

EllipSys3D solves the incompressible Navier–Stokes equations in general curvilinear coordinates using a finite volume method in multi-block structure Michelsen (1992, 1994); Sørensen (1995). Rhie-Chow interpolation is applied to prevent pressure decoupling, which is solved using an improved version of the SIMPLEC algorithm (Shen et al., 2003). The convective terms
are discretized using a fourth-order central difference scheme which includes an artificial viscosity term to suppress numerical wiggles (Wit and van Rhee, 2013), and time stepping is second-order with subiterations. Several RANS and LES turbulence models are implemented in EllipSys3D, where the anisotropic mimimal dissipation (AMD) Abkar et al. (2016) model has been utilized in the current simulations. Rayleigh damping is applied at high altitudes (> 1000m).

Initially, a precursor is simulated to spin up the CNBL. The precursor is performed in a domain  $L_x \times L_y \times L_z = 10,240$  m ×10,240 m ×3,000 m with a total of  $N_x \times N_y \times N_z = 512 \times 512 \times 384 \approx 100 \cdot 10^6$  cells corresponding to mesh resolution of  $\Delta x \times \Delta y \times \Delta z = 20$  m ×20 m ×5 m in the streamwise, lateral, and vertical direction. The equidistant mesh in the vertical is maintained at an altitude of 1,500m after which the cells are stretched. Cyclic boundary conditions are imposed in the streamwise and lateral direction, while a wind direction controller is imposed to continuously adjust the wind direction at z = 150 m to ensure that the flow direction is aligned with the wind turbines at hub height (Sescu and Meneveau, 2014;

Allaerts and Meyers, 2015). The precursor is initially spun up for 20 hours after which cross-stream planes are extracted for a total duration of 2 hours.

Subsequently, a mesh is build for the successor, which is  $L_x \times L_y \times L_z = 30,000 \text{ m} \times 30,000 \text{ m} \times 3,000 \text{ m}$  with a total of  $N_x \times N_y \times N_z = 512 \times 448 \times 192 \approx 44 \cdot 10^6$  cells. The mesh has a central equidistant region of  $L_{x,equi} \times L_{x,equi} \times L_{x,equi} = 13,530 \text{ m} \times 12,120 \text{ m} \times 1,500$  with  $\Delta x \times \Delta y \times \Delta z = 30 \text{ m} \times 30 \text{ m} \times 10$  m in the streamwise, lateral, and vertical direction with

- cells stretched to the boundaries. The precursor planes have been repeated to cover the extended domain of the successor simulations. The wind turbines are modeled by applying body forces in EllipSys3D, which is fully coupled to the aero-elastic tool HAWC2 Larsen and Hansen (2007) through the Dynamiks interface<sup>1</sup>. Velocities are transferred from EllipSys3D to HAWC2, which calculates aerodynamic forces and deflections, which are transferred back to EllipSys3D (Sørensen et al., 2015; Hodgson et al., 2022, 2023). HAWC2 also contains a dynamic torque controller, which enables the turbines to respond to the dynamic.
- ically changing inflow by dynamically updating pitch and rotational speed, but it does not yaw the turbines. The impact of realistic and dynamic wind turbine controllers has been shown to have a significant influence on power production for wind farms (Troldborg and Andersen, 2023a). Turbines can be modeled as actuator lines (Sørensen and Shen, 2002) or as actuator discs (Mikkelsen, 2004), which is used in this study. The simulations are run for 2 hours, where the initial 1 hour transient is discarded as the flow is still developing.

 $<sup>^{1}</sup> https://dynamiks.pages.windenergy.dtu.dk/dynamiks/index.html https://dynamiks/index.html https$