# Peer review of "An inter-comparison study on the impact of atmospheric boundary layer height on gigawatt-scale wind plant performance"

_Wind Energy Science, 2025_

## Author Comment (AC2)

**Authors Response to Review (reviewer 2)**

Ivanell et al.

October 24, 2025

The authors thank the reviewer for the time taken reviewing the paper and the helpful comments. This document specifies the modifications made according to the comments received by the reviewer. All changes are highlighted in a separate pdf for simplicity in parallel to the new (clean) version of the manuscript. In this document, we also comment on and discuss each review point made by the reviewer.

The input of the reviewer has improved the quality of the paper, we thank the reviewer for all contributions. In addition, the results from DTU have been updated due to the identification of a bug. Sincerely,

Authors

**Reviewer 2**

In this work, the authors investigated the impacts of atmospheric boundary layer (ABL) height and capping inversion layer thickness on wind plant performance. It is an important topic considering the continuing growth of wind plant sizes. One strong point of the work lies on the inclusions of results from different models and different codes. Specific comments are as follows:

1. The work focus on the power output and mean streamwise velocity. As the momentum entrainment in the vertical direction plays an important role on the wake flow recovery, and therefore the wind farm performance. It is necessary to examine turbulence statistics around the top tip of the rotor for different ABL heights and capping inversion layer thicknesses.

**Response#: The authors thank the reviewer for identifying the importance of the topic. We do agree with the reviewer that the entrainment is a very important aspect of wind farm flows. However, that has not been the focus of this paper, where the primary focus is to investigate how the farm efficiency depends on the boundary layer height. This is of course dependent of lateral and vertical entrainment. In this case, with the different level of boundary layer height, the vertical is of most interest. We refer, e.g., to Lanzilao Meyers (2024), where this type of analysis has already been thoroughly made. In the current study, we focus on model inter comparison rather and try to avoid repeating too much of this earlier work, so that the manuscript do not become overly long.**

2. There are several concerns regrading the conclusion, i.e., "the sensitivity of using different levels of modeling fidelity and numerical approaches overall is limited since the result generally show good agreement.", drawn in section 5: (1) the overall good agreement among the several codes employed in this work may not be enough to conclude that the sensitivity is limited;

(2) moreover, works in the literature have shown that recovery rates of wake flows depend on turbulence models and wind turbine models employed; and (3) it needs to be clear about for which quantities, the sensitivity is limited.

**#Response#:** We agree with the reviewer that the statement was too strong. This has been modified and moved to the discussion part instead of the conclusion.

3. Differences among different codes are larger for the H150 case and the H500 case in comparison with the H500-dh500 case (figures 6, 13-15). It is suggested to discuss the underlying reasons.

**Response#: Thank you for pointing this out. This is because DTU and UU did not simulate the H500-dh500 case, so there are fewer results to compare for this case. Consequently, the results in Figure 6 and 15 are less different compared to those in Figures 4, 5, 13 and 14. The additional comment has been added in the manuscript as follows: "Since DTU and UU did not simulate the H500-dh500 case, fewer results are available for comparison, which makes the results in Figure 15 appear less different than those in Figures 13 and 14.".**

4. It is necessary to use the same naming convention for legends of different codes. For instance, "DNV-RANS, DTU-LES, ..." are employed in figure 12, while "NDV, DTU, ..." are employed in figure 13. It is suggested to use those in figure 12.

**#Response#:** The legends have been corrected as suggested by the reviewer.

5. For the caption of each figure, make it self-explained. For instance, it is suggested to add the descriptions for each subfigure in figure 2.

**Response#: Labels in each subfigure in Figure 2 have been added.**

6. A typo on line 258: "caes"

**#Response#:** This has been corrected to "cases".

---

## Author Comment (AC3)

**Authors Response to Review (reviewer 1)**

Ivanell et al.

October 24, 2025

The authors thank the reviewer for the time taken reviewing the paper and the helpful comments. This document specifies the modifications made according to the comments received by the reviewer. All changes are highlighted in a separate pdf for simplicity in parallel to the new (clean) version of the manuscript. In this document, we also comment on and discuss each review point made by the reviewer.

The input of the reviewer has improved the quality of the paper, we thank the reviewer for all contributions. In addition, the results from DTU have been updated due to the identification of a bug.

Sincerely

Authors

**Reviewer 1**

This is a timely and ambitious paper that addresses the impact of atmospheric boundary layer (ABL) height and inversion thickness on the performance of large wind farms. The intercomparison of five numerical solvers, spanning both LES and RANS approaches, is a valuable contribution to the field. The paper is well-structured, and the methodology is described in sufficient detail to allow reproduction. The authors are commended for their transparency in discussing model differences and limitations.

The paper has the potential to make a significant contribution to our understanding of ABL effects on wind farm performance. However, the authors should take care to distinguish between physical insight and model sensitivity, and to communicate the implications of their findings more clearly. With a major revision, the paper could be a valuable reference for both researchers and practitioners in wind energy.

**#Response#:** We thank the reviewer for identifying this paper as a potential significant contribution. We agree that we can be more clear and specific about the separation of physical insight and modeling sensitivity. We have modified the discussion section to better describe this.

The paper raises several concerns that should be addressed before publication. Most notably, the reported wind farm efficiencies (60%) are significantly lower than typical assumptions (85%) for similar conditions. While this may reflect a real and underappreciated physical effect, the lack of standardized inflow conditions and observational validation makes it difficult to separate physical insight from model artifacts. There is a risk that these results, if not carefully contextualized, could undermine confidence in high-fidelity modeling for wind resource

assessment (WRA). Please include a discussion comparing your efficiency results with typical values used in WRA (e.g., 80–85%), and clarify whether the discrepancy is due to ABL effects or modeling setup. Consider adding a paragraph in the Discussion section explicitly addressing how these results should be interpreted by practitioners — i.e., not as a failure of LES/RANS, but as a call for better standardization and validation. Also, could this explain part of the overprediction bias in wind farm production? Similarly, the effect of blockage expressed as non-local efficiency appears to be much larger than even the most pessimistic estimates from the operational wind farm performance calculations.

**Response#: We think there might be a misunderstanding here and we also clarified to avoid that in the paper, see discussion section. We do not agree that this is significantly lower than in literature. However, we should be more clear on the fact that this is not the integrated efficiency considering a wide range of cases covering different velocities, wind directions, potential temperature profiles, etc. Instead, this case corresponds to an undisturbed wind speed close to rated, a relatively dense farm, with an inflow parallel to the rows of turbines. Comparable results from e.g. Lanzilao Meyers (2024, 2025) show similar levels of farm efficiency.**

In this work we did spend great effort on achieving similar inflow conditions despite the complexity of doing that with a range of models. However, the reviewer raises a good point on standardized inflow conditions in the literature for easier comparison.

The discussion section has been extended to include a discussion on how these results can be used in general.

Regarding the level of non-local efficiency, we believe this also depend on the specific cases studied here, i.e., that this is not integrated values for a wider set of flow cases.

The absence of observational validation is a limitation. Even a qualitative comparison with met mast or LiDAR data would strengthen the conclusions. If such data are unavailable, please state this explicitly and suggest how future work could address this gap.

**Response#: The reviewer raises a very important point for the entire community. Validation of flows around wind farms with 100 15 MW turbines is lacking. And measurement campaigns to map the flow would be a major effort. However, the inflow conditions can be mapped. In this study, we followed the suggested cases by Lanzilao and Meyers [1] for inflow that are well established reference cases, and probably the best cases that are available to date. A comment has been made to the manuscript explicitly explaining the lack of validation data, and the need for extensive validation campaigns.**

Several figures are placed far from their textual references, which makes the paper harder to follow. Consider reordering or duplicating key figures closer to where they are discussed.

**Response#: We have updated the figures position to be able to follow the paper more easily**

The inflow profiles differ substantially between solvers, especially in the H150 case. While the authors acknowledge this, the implications for the efficiency metrics  $(\eta_w, \eta_{nl}, \eta_f)$  are not fully explored. Since these metrics assume identical inflow, the comparison across solvers is

questionable. Could you quantify how much of the efficiency variation is attributable to inflow differences?

**Response#: As acknowledged in the manuscript, the H150 inflow profiles from e.g. DNV are outliers relative to the profiles from the other contributors. The manuscript goes on to suggest that these outlier profiles result in outlier results from the DNV model for the H150 case. We decided to test this by having DNV rerun the H150 case with inflow profiles more consistent with those of the other participants. This involved constructing an average potential temperature profile from the inflow profiles of the other participants. DNV ran this new potential temperature profile through its precursor model to create a new set of inflow profiles, which were used in a rerun of the H150 case. These new profiles are shown back-to-back with the original DNV inflow profiles in Figure 1. The new DNV wind speed and wind direction profiles are more consistent with those of the other participants. The new DNV potential temperature profile is also more consistent with the others (by design). The new DNV TKE profile is arguably less consistent with those of the other contributors, but it is not an outlier.**

Figure 1: Inflow profiles of the H150 case.

When the DNV model is run with the new inflow profiles, the results closely matches with the others as shown in the average streamwise velocity in Figure 2.

The new DNV simulation yields a higher non-local efficiency and total farm efficiency, however there is a slight drop in the wake efficiency.

|           | $\eta_{ m nl}$ | $\eta_{ m w}$ | $\eta_{ m f}$ |
|-----------|----------------|---------------|---------------|
| KUL       | 0.704          | 0.675         | 0.475         |
| DNV       | 0.656          | 0.627         | 0.412         |
| DNV-rerun | 0.695          | 0.620         | 0.431         |
| EDF       | 0.685          | 0.630         | 0.431         |
| UU        | 0.714          | 0.656         | 0.469         |
| DTU       | 0.768          | 0.652         | 0.501         |

Table 1: Wind Farm Efficiencies for the H150 Case

In the manuscript, we are sticking with the original DNV results, as these best reflect the full DNV approach, including precursor simulations, for simulating wind farm flows.

Another source of differences between the solvers seem to be how the wind turbines are represented. In Star-CCM+ for example, the forces are derived from the power and Ct curves,

Figure 2: Mean streamwise velocity averaged over the wind farm width at the hub height.

while they are calculated more explicitly in the EllipSys3D setup. Please comment. In addition, it would be important to know how the flow structure in terms of wind shear and wind veer may affect the power calculation in different solvers. For example, given a power curve, the wind turbine power is then usually calculated only using the hub-height wind speed, while the real power conversion also depends on the level of turbulence, wind profile, wind veer.

**Response#: Thank you for pointing out this issue. The following comment has been added in the discussion section. "Another source of difference between the simulation tools is the turbine representation, as described in Section 2. The turbine-induced aerodynamic forces differ, as KUL and EDF used a non-rotating disk model, while DNV, DTU and UU used a rotating disk model. It is difficult to isolate the impact of this discrepancy in the current set of simulations, as other factors, such as differences in inflow conditions and mesh resolutions, are also involved. Some previous studies have demonstrated that the choice between non-rotating and rotating disk methods may only affect the near-wake profile, while both models yield good agreement in the far wake (Meyers and Meneveau, 2010; Wu and Porté-Agel, 2011; van der Laan et al., 2015). Nonetheless, the impact of turbine-induced force distribution on wind farm efficiency, particularly a dense wind farm, has yet to be verified. Furthermore, the turbine controllers also differ among the tools: EDF and KUL used a constant thrust coefficient, DNV and UU used an averaged disk velocity to reference tabulated data, and DTU simulated a full controller within an aeroelastic code. These differences cause the turbines to respond differently to the incoming flows. For instance, DTU's full controller is capable of simulating a more realistic turbine response under varying wind shear and wind veer in the H150 and H500 cases, as the variable speed and pitch are controlled using the instantaneous torque on the disks as the input signal. However, the simpler controllers used by DNV and UU may underestimate the impact of varying velocity shear and directional veer on the turbine power output due to the averaging of rotor disk velocity. This issue is particularly significant for the large rotor size of the 15 MW turbine, as it can cause the turbines to respond differently to the incoming flows, and using a static or dynamic controller can significantly impact the estimated power production (Troldborg and Andersen, 2023). Nevertheless, although it is important to acknowledge these differences in turbine models, we would like to emphasize that the single turbine simulations that were used to normalize results (e.g. in terms of non-local and wake efficiency) partly factor out these differences."**

**L22 Stull 88 would be a better reference for the definition of a capping inversion.**

**Response#: The reference for the capping inversion has been changed to Stull (1988).**

L38 The GW discussion could profit from a calculation of the Froude number or an equivalent measure to determine the criticality of the flow over the WF. It would be for example interesting to see what happens in case when the inflow velocity is changed so that the flow at the WF is exactly critical.

**Response#: In contrast to classical shallow-water flow, the effect of the Froude number (sub versus supercritical) is less dominant here because of the internal waves, that always have an upstream footprint – from that perspective the Fr = 1 case does not lead to a singularity in the linearized perturbation equations. Nevertheless, we added the Fr and  $P_N$  numbers (relevant for waves on the capping inversion and internal waves, respectively) to the text before table 1**

L42 "adverse pressure gradient" needs some explanation.

**Response#: A clarification has been added to the manuscript.**

L64 It would be easier for the reader if this information was provided after Table 2.

**Response#: The information "It should be noted that DTU and Uppsala UU did not perform the simulation for the H500-dh500 case." has been moved to provide the detail after Table 2.**

L66 Please explain how the staggering was chosen. Is it to achieve minimal wake loss for the 270 degree wind?

**Response#: The case was chosen as a more relevant and realistic case considering the dominant wind direction in a wind farm, and avoiding the exactly aligned case that has been studied a lot in the past, but is not very representative for dominant wind directions. In general, there is a lack of reference wind farms and this layout will be suggested as one potential reference case to the community for the future.**

L73, Figure 1. If possible, please simulate the 315 degree flow (or rotate the WF by 45 degrees) and analyze how that affects the non-local efficiency (blockage).

**#Response#:** That is an interesting suggestion, however, the time of the project, and associated computer resources, are too limited to add these simulations at this point.

L195 "horizontal driving pressure gradient" is a sensitive subject in context of atmospheric CFD modelling attempts (in parallel with the Coriolis force), and requires more explanation in the paper. Has the flow achieved geostrophic balance during the precursor simulations (in all of the solvers) and the pressure gradient is then manipulated/adjusted during the production run to keep the winds aligned? Including a diagram of the model setup including the imposed pressure gradient would be best.

**Response#: The precursor runs in SP-Wind etc. are initialized using the geostrophic balance, and**

during the spin-up this balance is retained in the free atmosphere. A wind-angle controller is used to smoothly rotate the coordinate system during this period to keep the wind-direction at hub height oriented in the x-direction. During the production run (using the concurrent precursor method), the main domain is simply forced with turbulent inflow from the precursor - there is no additional manipulation necessary.

Due to the limitation in the wind angle controller in the SOWFA code, the desired geostrophic wind vector and the hub height wind direction cannot be achieved simultaneously during the precursor simulations. The controller in SOWFA is designed to control the mean wind speed and direction at a particular height which is practical for turbine engineering analyses. Nonetheless, compared to other codes, SOWFA gives a good agreement in the wind profiles across the turbine height. We added the discussion as follows: "This is done by determining the source term from the error between the actual planar-averaged velocity at the specified height and the desired velocity. This approach is suitable for turbine engineering analyses where the hub height wind speed can be prescribed. However, it is acknowledged that this method cannot simultaneously achieve both the desired geostrophic wind vector and the hub height wind direction."

L232, Figure 2. The U and V velocity components look OK and indicate westerly wind (from 270 degrees) at the hub height, and the wind above the BL has a negative V component and is from WNW, which is correct (on the N hemisphere). But the wind direction plot, (d), shows 0 at hub height and negative direction change above the BL which would mean WSW. Please resolve this inconsistency and/or correct the plot accordingly.

**Response#: The wind direction shown in Figure 2d is defined by the angle between the wind vector and the x-axis, with coordinates referenced from Figure 1. The wind angle is considered positive when the wind vector points toward the positive x and y axes, and negative when it points toward the positive x and negative y axes. We have added the definition  $\arctan \frac{v}{u}$  in the subcaption of Figure 2d. We also added the description in the text as follows: "denotes the wind angle between the wind vector and the x-axis".**

Figure 8. The horizontal wind velocity for the H500 case differs prominently from the H150 case (Figure 7). Most notably, the horizontal flow divergence is much stronger, and wavefronts are visible, indicating that the flow is supercritical. Please calculate or estimate the flow regime.

**#Response#:** This has been added to table 1.

Figure 13. Claiming that the non-local (blockage) effects account for more than 20% power loss already in the first row of turbines, even in the more optimistic H500 case, is bold and significantly deviates from traditional assumptions, and will raise eyebrows. In the worse case this may lead to loss of confidence in the concerned modelling approach. Please see the general comment as well, and it would be recommended to present this aspect of the paper carefully.

**#Response#:** We agree with the reviewer that this in not according to traditional assumed levels and not even common. We have added a comment on this in the manuscript, highlighting that this is the result of this specific case.

L298 The transparency is appreciated, i.e. the purpose of the study may be to only illustrate

the impact of the BL height, but the numeric results open important questions and call for revision of the traditional wind farm performance evaluations, and it may be questioned if just being open about the partial intent is sufficient for qualifying this paper for publication.

**Response#: We would like to return to the point that the specific cases performed here is not a representation of the average performance and therefore do not question the traditional wind farm performance. It rather uses one specific flow direction case. We have clarified this in the new version.**

L309 Are we sure that the waves in Figure 18 are non-physical? Internal GW may propagate in a stratified fluid like this.

**Response#: The waves above the wind farm may be considered as an internal gravity waves triggered by the turbines in a stratified ABL. However, there are non-physical waves trapped near the inlet. We have discussed this in the text "... resulting in reflection waves trapped near the inlet for both BLH cases that were not completely eliminated, ..."**

L319 Conclusions do not mention the huge contributions of internal and blockage effects to wind farm efficiency, which deserve some discussion.

**Response#: We have added a discussion on this and how representative this is for typical cases.**

Appendix A.  $U_d$  and gamma are not introduced. Please introduce them.

**Response#: The definitions for  $U_d$  and  $\gamma$  have been added in the Appendix A.**

**References**

[1] L. Lanzilao and J. Meyers. A parametric large-eddy simulation study of wind-farm blockage and gravity waves in conventionally neutral boundary layers. *Journal of Fluid Mechanics*, 979:A54, 2024.